# Complement System and Adhesion Molecule Skirmishes in Fabry Disease: Insights into Pathogenesis and Disease Mechanisms

**DOI:** 10.3390/ijms252212252

**Published:** 2024-11-14

**Authors:** Albert Frank Magnusen, Manoj Kumar Pandey

**Affiliations:** 1Division of Human Genetics, Cincinnati Children’s Hospital Medical Center, Cincinnati, OH 45229, USA; albert.magnusen@cchmc.org; 2Department of Pediatrics, College of Medicine, University of Cincinnati, Cincinnati, OH 45229, USA

**Keywords:** inflammatory cascade, endothelial dysfunction, cell adhesion molecules, complement-mediated injury, immune cell infiltration

## Abstract

Fabry disease is a rare X-linked lysosomal storage disorder caused by mutations in the galactosidase alpha (*GLA*) gene, resulting in the accumulation of globotriaosylceramide (Gb3) and its deacetylated form, globotriaosylsphingosine (Lyso-Gb3) in various tissues and fluids throughout the body. This pathological accumulation triggers a cascade of processes involving immune dysregulation and complement system activation. Elevated levels of complement 3a (C3a), C5a, and their precursor C3 are observed in the plasma, serum, and tissues of patients with Fabry disease, correlating with significant endothelial cell abnormalities and vascular dysfunction. This review elucidates how the complement system, particularly through the activation of C3a and C5a, exacerbates disease pathology. The activation of these pathways leads to the upregulation of adhesion molecules, including vascular cell adhesion molecule 1 (VCAM1), intercellular adhesion molecule 1 (ICAM1), platelet and endothelial cell adhesion molecule 1 (PECAM1), and complement receptor 3 (CR3) on leukocytes and endothelial cells. This upregulation promotes the excessive recruitment of leukocytes, which in turn exacerbates disease pathology. Targeting complement components C3a, C5a, or their respective receptors, C3aR (C3a receptor) and C5aR1 (C5a receptor 1), could potentially reduce inflammation, mitigate tissue damage, and improve clinical outcomes for individuals with Fabry disease.

## 1. Introduction

Fabry disease is a rare X-linked lysosomal storage disorder with an estimated prevalence of 1 in 20,000 to 60,000 live births. However, newborn screening data suggest a potentially higher prevalence, with estimates around 1 in 8800 [1,2]. The disease is caused by mutations in the galactosidase alpha (*GLA*) gene, which encodes the enzyme alpha-galactosidase A (α-Gal A) [3,4,5,6,7,8]. This enzyme plays a vital role in the breakdown of globotriaosylceramide (Gb3 or GL-3) that tends to accumulate in various cells and tissues and globotriaosylsphingosine (Lyso-Gb3), a water-soluble compound that found in body fluids, (e.g., serum and plasma) [9,10,11,12,13,14,15,16,17,18]. Deficient α-Gal A activity leads to the pathological accumulation of Gb3 and Lyso-Gb3 in multiple cell types, including peripheral blood mononuclear cells, renal cells (such as epithelial, endothelial, and podocytes), vascular smooth muscle cells, cardiomyocytes, and neurons, and in body fluids like serum, plasma, and cerebrospinal fluid [9,10,11,12,13,14,15,16,17,19,20,21,22].

The abnormal accumulation of Gb3 and Lyso-Gb3 disrupts cellular functions and causes tissue damage across multiple organs, including the liver, spleen, kidneys, skin, blood vessels, and peripheral nerves [23,24,25]. Early symptoms of Fabry disease include episodes of pain (acroparesthesia), reduced sweating (hypohidrosis), gastrointestinal disturbances, and angiokeratomas [26,27,28,29,30]. As the disease progresses, it can lead to severe complications such as renal failure, cardiovascular disease, stroke, and neurodegenerative disorders [9,10,11,12,13,14,15,16,17,31,32,33,34,35,36,37,38,39,40]. The severity and progression of symptoms vary, with males generally exhibiting more severe manifestations compared to females, who experience variable symptoms due to random X-chromosome inactivation [41,42,43].

Current treatments, such as enzyme replacement therapies with agalsidase α or agalsidase β and chaperone therapy with migalastat, provide some symptom relief but are costly and do not halt the disease progression. These therapies are also associated with challenges like infusion-related reactions and the development of neutralizing immunoglobulin G (IgG) antibodies [44,45,46,47,48,49,50,51,52,53,54,55,56,57]. These limitations underscore the need for a deeper understanding of the Gb3/ Lyso-Gb3-induced disease mechanisms in Fabry disease.

Studies have increasingly highlighted the impact of immune dysregulation in Fabry disease, with a particular focus on the complement system. Research has consistently demonstrated elevated levels of complement components, including complement 1q subcomponent c (C1qc), C3, inactivated C3b (iC3b), C4, and complement factor B (CFB), alongside the production of complement fragments C3a and C5a, in serum, plasma, and tissue samples from both animal models and human patients with Fabry disease (Table 1).

The activation of C3aR by C3a and C5aR by C5a leads to the upregulation of key adhesion molecules, including complement receptor 3 (CR3), vascular cell adhesion molecule 1 (VCAM1), intercellular adhesion molecule 1 (ICAM1), and platelet and endothelial cell adhesion molecule 1 (PECAM1) on leukocytes and endothelial cells that cause excess tissue recruitment of macrophages, T cells, and B cells [60,61,62,63,64]. Such C3a–C3aR and C5a–C5aR axes mediated enhanced tissue recruitment of innate and adaptive immune cells contributes to progressive tissue damage in various conditions, including lung injuries, fibrosis, renal failure, cardiomegaly, stroke, and neurodegenerative diseases [65,66,67,68,69,70,71,72].

The complement system is an essential component of immune defense, designed to protect against infections and clear out damaged cells [73,74,75]. In Fabry disease, however, this system appears to be improperly activated. This dysregulation triggers a chain reaction of inflammatory responses that worsens the disease progression. Notably, there are increased levels of complement components and their active fragments, C3a and C5a [21,58,76] as well as enhanced expression of adhesion molecules on leukocytes and endothelial cells [10,11,77,78,79].

Furthermore, patients with Fabry disease exhibit elevated blood lymphocyte counts, increased levels of specific T cytotoxic cell subsets (CCR4^+^CXCR3^+^ and CCR6^+^), higher numbers of MHCII^+^ CD1d^+^ CD11b^+^ CD31^+^ monocytes, and notable tissue infiltration by macrophages and B cells [11,78,80,81,82,83,84,85,86]. These findings suggest that not only is the complement system excessively activated, but it also plays a role in aggravating the disease through increased endothelial cell abnormalities and enhanced tissue recruitment of immune cells.

Understanding the pivotal roles of C3a and C5a and their impact on the upregulation of adhesion molecules offers critical insights into the pathogenesis and progression of Fabry disease. This knowledge elucidates how Fabry disease impacts excessive tissue infiltration of innate and adaptive immune cells, leading to progressive organ damage. By targeting these pathways, we can explore novel therapeutic strategies that specifically address the inflammatory processes underpinning Fabry disease. Targeting the C3a–C3aR and C5a–C5aR1 pathways and their influence on adhesion molecule-mediated immune cell infiltration holds significant promise for mitigating inflammation and potentially halting or slowing the progression of organ damage in Fabry disease.

## 2. Complement Activation Pathways

The complement system, comprising approximately 40–50 proteins predominantly synthesized in the liver or locally by various immune and non-immune cells such as monocytes, macrophages, dendritic cells, lymphocytes, adipocytes, fibroblasts, and epithelial cells, plays a crucial role in immune defense [87,88,89,90,91,92]. They work in a highly regulated manner to defend against pathogens, clear immune complexes, and facilitate inflammation, ensuring the complement system operates effectively without causing excessive damage to host tissues [93,94,95,96,97]. The complement system encompasses diverse proteins categorized based on their functions. One of its primary roles is complement activation, achieved through three distinct pathways: the classical pathway, the alternative pathway, and the lectin pathway. These pathways employ specific recognition mechanisms and activation steps that synergistically enhance immune responses against pathogens, damaged tissues, and aid in the clearance of cellular debris and immune complexes. Each pathway contributes uniquely to the overall effectiveness of the immune response, ensuring comprehensive defense and the maintenance of immune balance.

The classical pathway is initiated when the C1 complex (composed of C1q, C1r, and C1s) binds to the fragment crystallizable (Fc) region of immunoglobulin M (IgM) or IgG antibodies attached to antigens or released from damaged cells. This interaction activates the serine proteases C1r and C1s, which cleave C4 and C2 proteins to form C4b2a, also known as C3 convertase, which cleaves C3 into C3a and C3b (Figure 1a). The C3b subsequently binds with C4bC2a to form C5 convertase (C4bC2aC3b), which cleaves C5 into C5a and C5b (Figure 1b). The C5b initiates the assembly of the membrane attack complex (MAC), consisting of C6, C7, C8, and multiple C9 molecules. The MAC forms transmembrane channels in microbial and host cell membranes, leading to cell lysis and the destruction of targeted microorganisms or host cells (Figure 1c).

The lectin pathway is initiated when mannose-binding lectin (MBL) or ficolins bind to carbohydrate patterns associated with pathogens or altered self-surfaces on host cells. This binding activates MBL-associated serine proteases (MASPs), as illustrated in Figure 1d. These MASPs subsequently follow the classical complement pathway, leading to the production of C3a and C5a, as shown in Figure 1a–c. Specifically, MASPs cleave C4 and C2 to form the complex C4b2a. C4b2a then cleaves C3 into C3a and C3b. The binding of C3b with C4b2a forms the C5 convertase (C4b2aC3b), which cleaves C5 into C5a and C5b. C5b subsequently initiates the assembly of the membrane attack complex (MAC), composed of C6, C7, C8, and multiple C9 molecules. The MAC forms a transmembrane channel in the membranes of both pathogens and host cells, resulting in cell lysis and the destruction of targeted microorganisms or damaged host cells.

The alternative pathway of complement activation is constitutively active at a basal level and amplifies upon the detection of pathogens or injured tissues. It begins with the spontaneous hydrolysis of C3, producing C3 (H_2_O), which then binds complement factor B (CFB). Factor D (FD) cleaves factor B to produce C3 (H_2_O)Bb, functioning as a C3 convertase, as shown in Figure 1e. This C3 convertase then follows a pathway similar to that of the classical pathways, cleaving C3 into C3a and C3b. C3b binds to C3 (H_2_O)Bb to form the C5 convertase (C3bBb), which subsequently cleaves C5 into C5a and C5b. This initiates the assembly of the MAC, comprising C6, C7, C8, and multiple C9 molecules. The MAC forms a transmembrane channel in both microbial and host cell membranes, leading to cell lysis and the destruction of targeted microorganisms or damaged host cells, as illustrated in Figure 1a–c.

All three complement activation pathways converge at the formation of C3 convertase (either C4b2a from the classical and lectin pathways or C3bBb from the alternative pathway). This enzyme cleaves C3 into C3a and C3b. C3b ligation to C4b2a or C3bBb causes the formation of C5 convertase, which cleaves C5 into C5a and C5b [90,91,92].

## 3. Complement 3a

C3a is a small peptide, approximately 9 kDa in size, composed of 77 amino acids arranged into four anti-parallel helical structures stabilized by three disulfide bridges. This structural configuration is crucial for its biological activity and interaction with its receptor [98]. Upon binding C3aR, a 55 kDa G-protein coupled receptor (GPCR) with a distinctive seven-transmembrane domain, C3a exerts its effector functions. This receptor is instrumental in mediating the effects of C3a across various physiological and pathological contexts [98,99,100,101]. C3aR is widely expressed in numerous organs, including the heart, kidneys, brain, liver, lungs, intestines, and skeletal muscle, as well as in various cell types such as monocytes, macrophages, dendritic cells, neutrophils, eosinophils, basophils, mast cells, vascular endothelial cells, choroid plexus epithelium, neurons, microglial cells, astrocytes, oligodendrocytes, and Purkinje cells [99,101,102,103,104,105,106,107,108,109,110,111,112,113,114,115,116,117].

## 4. Complement 5a

C5a plays a fundamental role as an anaphylatoxin and chemoattractant in the immune system, contributing to inflammatory responses and host defense mechanisms. Structurally, C5a is a glycoprotein weighing approximately 15 kDa, composed of 74 amino acids [118]. C5a is cleaved from the amino terminus of the alpha chain of the much larger C5 protein, which is a 190 kDa protein consisting of an alpha chain (~120 kDa) and a beta chain (~75 kDa) connected by disulfide bonds [119]. C5a exerts its effects through interaction with two distinct receptors: C5aR1 (CD88) and C5aR2 (GPR77 or C5L2) [120,121,122]. C5aR1, a member of the G-protein coupled receptor (GPCR) family, is a 43 kDa protein with 350 amino acids [118,123,124]. C5aR1 expression has been identified in a diverse array of immune cells, including monocytes, macrophages, dendritic cells, neutrophils, mast cells, T cells, and B cells, as well as in non-immune cells such as endothelial cells, mesangial cells, hepatocytes, keratinocytes, and neuronal cells. This expression is also observed across various tissues, including the liver, spleen, lungs, and brain [125,126,127,128,129,130,131,132,133,134]. C5aR1 possesses a classic seven-transmembrane domain structure typical of GPCRs and contains specific binding sites that recognize C5a [135,136,137]. C5aR2 shares 58% sequence homology with C5aR1 and has a molecular weight of approximately 37 kDa [122]. It is widely expressed in various tissues, including the liver, spleen, lungs, kidneys, heart, testis, and adipose tissues, as well as in cells such as macrophages, dendritic cells, neutrophils, cardiomyocytes, and renal cells [121,122,138,139,140]. C5aR2 serves as a crucial non-signaling decoy receptor that binds to C5a, but it exhibits an even stronger affinity for C3a and acylation-stimulating protein (ASP/C3a des Arg) than C5aR1 does. By effectively sequestering C5a, C5aR2 regulates its extracellular bioavailability, significantly reducing the amount accessible to C5aR1. This mechanism helps to temper the pro-inflammatory response typically triggered by C5a, highlighting the crucial balance of immune regulation at play [140,141,142,143,144,145].

## 5. Complement Activation and the Production of Complement 3a and Complement 5a in Fabry Disease

Several complement components that play a key role in activating the complement pathways to produce C3a and C5a, such as C1qc, C3, C4, and CFB have been detected in the serum, plasma, and brain tissues of Fabry patients (Table 1). Specifically, a study by Laffer et al. found elevated serum levels of C3a and C5a in untreated male patients with Fabry disease carrying missense or nonsense mutations. Notably, C3a concentrations were found to be significantly higher, up to 30-fold, when compared to healthy controls, while C5a levels demonstrated a more modest 2-fold increase [21]. This discrepancy is likely due to the increased tissue expression of C5aR1 and C5aR2 compared to C3aR and/or differing biological characteristics of the degradation products C3adesArg and C5adesArg, with C5adesArg retaining its ability to bind to complement receptors, unlike its counterpart [140,141,142,143,144,145,146].

Aging is a well-established risk factor for a variety of conditions, including age-related macular degeneration, metabolic disorders, and neurodegenerative diseases [147,148,149], all of which are influenced by complement activation [150,151,152,153,154,155]. In this context, Fabry disease is particularly notable due to its association with reduced life expectancy and indications of premature aging [156,157]. This raises critical questions about the relationship between aging, the production of complement components C3a and C5a, and the progression of disease.

Interestingly, a study by Laffer et al. revealed that levels of C3a and C5a in patients with Fabry disease do not correlate with age. This finding suggests that the dysregulation of these complement components may persist across different age groups [21]. This stresses the need for further investigation into how complement activation contributes to disease mechanisms in the context of aging and Fabry disease.

Importantly, C3a concentrations above 5000 ng/mL were observed in all treatment-naive patients with Fabry disease, and this elevated level persisted even after enzyme replacement therapy, particularly in individuals with marked reduction in Lyso-Gb3 and positivity for drug specific IgG antibodies (Drug-IgG) [21]. These findings collectively suggest Gb3-dependent, but lyso-Gb3-independent complement activation is responsible for the production of C3a and C5a in Fabry disease.

Indeed, excessive deposition of certain lipids, such as triglycerides and cholesterol, has been associated with complement activation [158,159,160,161,162,163,164,165,166]. We have previously demonstrated a significant presence of C5a in glucosylceramide-storage cells, such as macrophages and dendritic cells. Furthermore, we have shown that glucosylceramide-specific IgG autoantibodies induce both local and systemic complement activation, leading to C5a generation in mouse models and human patients with Gaucher disease [167]. In the treatment of Fabry disease, enzyme replacement therapy using either agalsidase α or agalsidase β induced the development of Drug-IgG, particularly of the IgG1 and IgG4 subclasses [168,169,170].

Building on these critical findings, we propose a mechanistic framework for the pathogenesis of Fabry disease. The excess tissue accumulation of Gb3 may induce spontaneous C3 hydrolysis by the alternative pathway of complement activation resulting in C3 (H_2_O) formation (Figure 2a). The binding of C3 (H_2_O) to complement factor B (CFB) and its subsequent cleavage by factor D (FD) causes the generation of the C3 convertase; C3 (H_2_O)Bb. This enzyme complex cleaves additional C3 molecules into C3a and C3b; C3b generated from the initial cleavage binds to CFB, which is then cleaved by FD to form C3bBb, a potent C3 convertase, C3bBb (Figure 2b). This C3bBb leads to the downstream cleavage of C3 into C3a and C3b (Figure 2c). The binding of C3bBb to C3b causes the formation of C5 convertase, C3bBbC3b (Figure 2d), which cleaves C5 into C5a in Fabry disease (Figure 2e).

However, post enzyme replacement therapy can lead to the formation of drug, i.e., agalsidase α- or agalsidase β-specific IgG antibodies (Drug-IgG), which bind to the indicated therapeutic drugs and forms the drug-specific IgG immune complexes (Drug-IgG ICs) (Figure 2f). The subsequent binding of C1qc to Drug-IgG ICs activates serine proteases C1r and C1s (Figure 2g). This activation triggers the cleavage of C4 and C2, ultimately forming C3 convertase (C4b2a) through the classical complement pathway (Figure 2h). The resulting C3 convertase (C4b2a), from classical pathways facilitates the downstream cleavage of C3 into C3a and C3b (Figure 2i). The interaction of C3 convertase with C3b subsequently leads to the formation of C5 convertase, which cleaves C5 into C5a in the context of Fabry disease (Figure 2j).

Key inquiries remain regarding the dynamics of complement activation in Fabry disease: Is this activation sporadic or continuous? To what extent is complement activation directly influenced by the accumulation of Gb3/ Drug-IgG ICs in tissues? Furthermore, are certain tissues more critical than others for mediating systemic effects? Understanding these relationships is essential for elucidating the progression of disease.

## 6. Role of Complement 3a and Complement 5a in Immune Cell Infiltration in Fabry Disease

The leukocyte recruitment of tissues is an essential step in the inflammatory response that requires the binding and extravasation of leukocytes in the vasculature. In tissues, it starts with the rolling of the leukocyte over the activated endothelium, followed by firm adhesion, diapedesis through the endothelial layer and further migration into the tissue matrix [171,172,173]. Adhesion molecules are cell surface proteins involved in mediating interactions between cells, as well as between cells and the extracellular matrix. There are four families of adhesion molecules: (1) leukocyte cell adhesion integrins, (2) immunoglobulin-like adhesion molecules, (3) cadherins, and (4) selectins, i.e., P, E, and L selectins, each playing distinct roles in the tissue recruitment of innate and adaptive immune cells [174,175,176,177,178,179,180,181,182,183,184,185,186,187,188]. Below is a detailed classification of these adhesion molecules.

### 6.1. Leukocyte Cell Adhesion Integrins

Leukocyte cell adhesion integrins are transmembrane receptors composed of α and β subunits that together form a functional receptor complex on the plasma membrane. These receptors bind to a diverse array of ligands, including components of the extracellular matrix, surface molecules on other cells, and soluble proteins, thereby mediating cell–cell and cell–extracellular matrix interactions. Integrins are crucial for processes such as immune cell adhesion and migration [189,190].

Leukocytes express a variety of integrins essential for their functions. These include β1-integrins such as Very Late Antigen 3 (VLA3), also known as α3β1 (CD49c+CD29), and VLA4, also known as α4β1 (CD49d+CD29). β2 integrins encompass Lymphocyte Function-Associated Antigen 1 (LFA-1), also known as αLβ2 (CD11a paired with the β2 chain CD18), complement receptor 3 (CR3), also known as αMβ2 or Mac-1 (CD11b paired with the β2 chain CD18), complement receptor 4 (CR4), also known as αXβ2 (CD11c paired with the β2 chain CD18), and αDβ2 (CD11d paired with the β2 chain CD18). Additionally, β7-integrins include α4β7 and αEβ7 (CD103) [191,192,193,194,195,196,197,198,199,200,201].

#### 6.1.1. Very Late Antigen 3 (VLA 3)

Very Late Antigen 3 (VLA3), also known as α3β1 integrin, is a crucial transmembrane receptor composed of α3 (CD49c) and β1 (CD29) subunits [199,200,201]. Several studies have defined multiple ligands for α3β1 integrin, like fibronectin, collagen, laminin, (laminin 1 and laminin 5), entactin, and tetraspanins such as CD9, CD63, and CD151 [202,203,204,205,206,207,208,209,210,211,212,213,214]. Additionally, α3β1 is involved in the infiltration of T and B cell subsets into tissues and is essential for the function of podocytes, which cover the glomerular basement membrane and are integral to kidney filtration [206,207,215,216,217,218].

The integrin α3β1 serves as the principal cell-matrix adhesion receptor in podocytes, forming crucial connections with laminin 521 in the glomerular basement membrane through various adaptor proteins that anchor it to the intracellular actin cytoskeleton [216]. Deficiencies in either the α3 or β1 integrin subunits, or in their ligand laminin, can lead to severe conditions such as fatal proteinuria [219,220,221]. Moreover, the deletion of the tetraspanin CD151, which interacts with both α3β1 integrin and laminin β2, has been linked to the development of nephrotic syndrome [221,222].

In the context of Fabry disease models induced by chloroquine, a decrease in α3β1 integrin expression in podocytes correlates with renal dysfunction, highlighting its im-portance in maintaining podocyte integrity [223]. In patients with Fabry disease experiencing nephropathy, the majority show no glomerular deposition of complement cleavage products, indicating a potential absence of a direct complement-mediated injury in glomeruli. Both the mouse model of Fabry disease and a small subset of Fabry patients with glomerulopathy exhibit renal tissue deposition of C3, predominantly in the mesangial area (Table 1). This suggests the potential for localized complement activation, which may play a role in the kidney damage observed in Fabry disease [15,58,59,224]. Interestingly, while α3β1 integrin does not bind directly to complement components like C3, it can interact with invasion, a bacterial protein that disrupts the adhesion between laminin 5 and α3β1 [225]. This raises intriguing questions about the specific roles of C3-independent α3β1 function within the inflammatory milieu of Fabry disease and suggests that further investigation is needed to elucidate how these complement components may influence integrin function and podocyte health in this context.

#### 6.1.2. Very Late Antigen 4 (VLA 4)

Very Late Antigen 4 (VLA 4), also known as α4β1 or CD49d/CD29, is a critical integrin found on the surface of various immune cells, including lymphocytes, monocytes, and eosinophils. Its interaction with VCAM1 plays a crucial role in the immune response by mediating the transition of these cells from a state of rolling along the blood vessel walls to a firm adhesion [179]. This transition is fundamental for the effective migration of immune cells to sites of inflammation, featuring the importance of VLA4 in facilitating precise immune responses.

In patients with Fabry disease, studies have noted a modest increase in the expression of VLA 4 on peripheral monocytes, along with a significant elevation in circulatory levels of solubleVCAM1 [78]. Furthermore, Gb3 stimulation of endothelial cells has been shown to induce upregulation of VCAM1 in the context of Fabry disease [10]. Together, these findings underscore the importance of the VLA4/VCAM1 axis in the infiltration of immune cells into tissues, suggesting it plays a pivotal role in the pathophysiology of the Fabry disease. 

#### 6.1.3. Leukocyte Function-Associated Antigen 1 (LFA1)

Leukocyte Function-Associated Antigen 1 (LFA1), also known as αLβ2 or CD11a/CD18, is a fundamental integrin involved in the guidance and migration of nearly all leukocytes, including T cells, B cells, natural killer (NK) cells, monocytes, macrophages, and neutrophils. By binding to its ligand, ICAM1, LFA1 facilitates crucial interactions between immune cells and the endothelial layer of blood vessels, enabling effective tissue infiltration and immune surveillance [226,227,228,229,230,231,232].

Its role extends beyond migration; LFA1 is essential for modulating immune responses, with its presence influencing T cell activation, B cell activation, and the apoptosis of target cells [233,234]. In autoimmune models, high LFA1 expression on memory B cells highlights its role in their trafficking [235,236]. Moreover, myeloid cells like monocytes and macrophages, as well as neutrophils, rely on LFA1 and CR3 for movement within activated blood vessels, illustrating the adaptability of the LFA1–CR3 axis in immune cell navigation [237]. In patients with Fabry disease, a significant increase in the expression of CD18, the β subunit of LFA1, on peripheral monocytes has been observed, along with elevated levels of ICAM1 in the bloodstream. These findings underscore the crucial role of LFA1 in the pathology of Fabry disease and its influence on immune function [78].

#### 6.1.4. αMβ2

αMβ2, also known as CR3, Mac-1, and CD11b/CD18, is expressed on a variety of phagocytic cells, including monocytes, macrophages, dendritic cells, neutrophils, and eosinophils, as well as on minor subsets of B cells, T cells, and natural killer (NK) cells [238,239,240,241,242]. CR3 recognition by ICAM1, ICAM2, and the complement fragment iC3b are crucial for neutrophil functions such as phagocytosis, reactive oxygen species formation, the formation of neutrophil extracellular traps, apoptosis, and cytokine production [190,192,231,237,242,243,244,245]. In human umbilical vein endothelial cells, the deposition of iC3b significantly enhances the adhesion of neutrophils to the endothelium. This enhancement is notably reduced by monoclonal antibodies targeting CD11b/CD18, indicating that the CR3–iC3b axis is essential for promoting leukocyte adhesion [246]. Additionally, the C3b–iC3b axis appears to be activated early in vasculitis lesions, leading to the accumulation of CD11b^+^ and CD14^+^ monocytes [247]. In patients with Fabry disease, elevated plasma levels of iC3b and an increased expression of CR3 on peripheral blood monocytes have been observed, highlighting the role of this axis in the pathology of Fabry disease [58,78].

#### 6.1.5. αXβ2

αXβ2, also known as CR4 and CD11c/CD18, is predominantly expressed on monocyte-derived macrophages and monocyte-derived dendritic cells. CR4 mainly binds to ICAM 3, VCAM1, and various extracellular matrix proteins, thereby playing a significant role in the function and migration of these immune cells [248,249]. CR4 is closely related to CR3 as the entire CR4 α-chain (CD11c) shares 63% sequence homology to the CR3 α chain (CD11b), and therefore, considered as homologous adhesion receptors that are expressed on similar types of leukocytes and recognize similar ligands such as iC3b and fibrinogen [244,245,250,251]. CR4 plays a central role in regulating the anti-inflammatory function of macrophages [251]. A deficiency of CR4 results in the loss of the antifungal activity of macrophages by eliminating their recruitment and adhesion function [251] and disturbing dendritic cell recruitment to the infection site [252].

#### 6.1.6. αDβ2

αDβ2, also known as CD11d/CD18, is the most recently identified β2-integrin. It is expressed at low levels on circulating neutrophils and monocytes but is highly expressed on M1-like macrophages within inflamed tissues [198,253,254,255,256]. αDβ2 is highly homologous to integrin αMβ2 and αXβ2. It binds with ICAM-1, ICAM 3, and VCAM1 [231,253]. Studies using primary leukocytes from humans and mice, as well as transfected cell lines, indicate that αDβ2 binds to ICAM1, ICAM-3, VCAM1, and various extracellular matrix proteins [198,248,253,256,257]. This binding suggests that αDβ2 plays a significant role in immune cell adhesion and migration.

#### 6.1.7. α4β7

The α4β7 integrin, expressed on various leukocytes including T and B cells, is crucial for directing these cells to gut-associated lymphoid tissues and inflamed skin [258,259,260,261,262]. This integrin specifically interacts with VCAM1 and mucosal addressin cell adhesion molecule-1 (MAdCAM-1) [259,261,262,263,264]. In Fabry disease, elevated levels of VCAM1 have been detected in peripheral blood, and Gb3 stimulated endothelial cells [10,77,78,79]. These findings show the importance of the α4β7–VCAM1 axis in the disease, highlighting its relevance to gastrointestinal inflammation and the broader complications associated with Fabry disease [265,266,267].

#### 6.1.8. αEβ7

αEβ7 integrin, also known as CD103, forms a heterodimeric integrin, which is mainly expressed in lymphocytes of intestinal, lung, and skin epithelial tissues as well as in conventional dendritic cells of the mucosa and dermis [268,269,270,271]. αEβ7 integrin binds to E cadherin on epithelial cells, playing a crucial role in maintaining tissue integrity and facilitating the retention of immune cells within epithelial tissues [272]. The interaction between αEβ7 and E cadherin mediates lymphocyte attachment to intestinal and skin epithelial cells [268]. In Fabry disease, immune cell dynamics are significantly influenced by integrins, particularly those within the β1, β2, and β7 families. While research into β7 integrins in this context is still developing, the roles of some of the β1 and β2 integrins have been more thoroughly investigated.

The β2 integrin family, comprising CR3 (CD11b/CD18), LFA-1 (CD11a/CD18), αXβ2 (CD11c/CD18), and αDβ2 (CD11d/CD18), is fundamentally dependent on the β subunit (CD18) [273,274]. This β2 integrin family is crucial for various leukocytes, including monocytes, macrophages, neutrophils, and dendritic cells [275,276]. An increase in CD11b and/or CD18 expression has been linked to a range of inflammatory diseases, such as nephropathy, pulmonary disorders, hypertension, vascular dysfunction, neurological conditions, and immune complex-mediated diseases [60,277,278].

Conversely, defects in the synthesis of the common β-chain (CD18) lead to a rare autosomal-recessive condition known as leukocyte adhesion deficiency (LAD)-I. This disorder is characterized by the absence or diminished expression of crucial integrins on leukocytes, resulting in severe deficiencies in leukocyte adhesion, chemotaxis, and aggregation. Consequently, patients with LAD-I experience frequent bacterial infections, impaired wound healing, and often face a grim prognosis, frequently succumbing during childhood [279,280]. A related condition, LAD-III, shares a similar phenotype but is caused by mutations in Kindlin-3, a protein essential for the activation of β2-integrins through inside-out signaling. On the other hand, LAD-II, although it presents with similar immunodeficiency issues, does not involve direct defects in integrin function. Instead, it stems from mutations affecting a GDP-fucose transporter, which impairs the synthesis of selectin glycoprotein ligands, disrupting leukocyte rolling and ultimately hindering their extravasation [281].

Patients with Fabry disease display notable changes in integrin expression. Specifically, there is a slight increase in VLA 4 (α4β1 integrin) on peripheral monocytes, suggesting a role in their altered behavior. More pronounced are the findings related to β2 integrins: Fabry patients show elevated levels of iC3b in circulation and a heightened expression of CR3/Mac1, an integrin complex comprising α (CD11b) and β (CD18) subunits, on peripheral blood monocytes [58,78]. The upregulation of CR3 and the increased presence of macrophages in endomyocardial tissues from patients with Fabry disease suggest the crucial role of immune cell migration in the progression of disease [78,84].

In our previous study, we demonstrated that the C5a–C5aR1 axis mediates the upregulation of CR3, which plays a critical role in the excessive infiltration of macrophages and neutrophils into tissues [60]. Conversely, inhibiting C5aR1 effector functions significantly reduced this infiltration, suggesting the importance of the C5a–C5aR1 axis in regulating immune cell migration [60,61]. In a *Gba1*-prone mouse model of (Gba1 ^9V/−^) and conduritol B epoxide (CBE)-mediated glucocerebrosidase-targeted experimental mouse model of Gaucher disease, we have shown an increased recruitment of CR3-expressing macrophages and dendritic cells to tissues including the liver, spleen, and lungs. This recruitment was markedly diminished in both the C5aR1-deficient Gba1 ^9V/-^ and the C5aR1-deficient CBE-mediated glucocerebrosidase-targeted mouse models of Gaucher disease, emphasizing the role of C5aR1 in this process [167].

These findings suggest that the C5a–C5aR1 axis, through its influence on CR3 ex-pression and interactions with endothelial cells and integrins such as VLA4 and LFA1, may contribute to the excessive recruitment of leukocytes observed in Fabry disease. This mechanism is particularly relevant to the increased infiltration of macrophage, T cell, NK cell, and B cell precursors into the bloodstream and tissues in patients with Fabry disease [11,82,84,85,86].

Therefore, interaction between the C5a–C5aR1 axis and its impact on CR3, along with other β2 integrins, provides valuable insights into the altered immune dynamics, warranting further investigation into the pathophysiology and effects on immune cell function. A deeper understanding of these interactions may reveal potential pathways for developing novel therapeutic strategies aimed at modulating tissue inflammation in Fabry disease.

### 6.2. Immunoglobulin-like Adhesion Molecules in Fabry Disease

Immunoglobulin-like adhesion molecules, such as VCAM1, ICAM1, and PECAM1, are critical components in the regulation of cell adhesion and immune cell trafficking. These molecules are integral to maintaining cellular interactions within various tissues and play pivotal roles in cell adhesion and immune cell trafficking [174,175,282,283,284,285,286].

#### 6.2.1. Vascular Cell Adhesion Molecule 1 (VCAM1)

Vascular cell adhesion molecule 1 (VCAM1), also known as CD106, is a pivotal 90 kDa glycoprotein integral to immune cell adhesion and migration [285,286]. VCAM1 is primarily inducible and expressed on endothelial cells, where it plays a critical role in inflammation and immune responses [287,288]. VCAM1 influence extends beyond just endothelial cells. It is also found on various immune cells, such as macrophages and dendritic cells, as well as non-immune cells including bone marrow fibroblasts, myoblasts, oocytes, Kupffer cells, Sertoli cells, and even certain cancer cells [283,284]. This widespread expression underscores the versatility and importance of VCAM1 in cellular interactions. VCAM1 interaction with galectin-3 on eosinophils, and with VLA4 and α4β7 on lymphocytes, monocytes, and macrophages, is crucial for facilitating the firm adhesion of indicated leukocyte to the endothelium [179,285,286,289,290,291]. This interaction enables these cells to migrate efficiently into inflamed tissues, underlining the essential role of VCAM1 in orchestrating immune responses and tissue inflammation.

#### 6.2.2. Intercellular Adhesion Molecule 1 (ICAM1)

Intercellular adhesion molecule 1 (ICAM1), also known as CD54, is a crucial cell surface glycoprotein with a molecular weight ranging from 60 to 114 kDa, which is prominently expressed on endothelial cells, antigen-presenting cells, and various other cell types [292,293]. ICAM-1 plays a pivotal role in immune responses by interacting with LFA1 and CR3 on T cells, macrophages, and neutrophils. These interactions are essential for facilitating cell adhesion and transmigration, processes that are critical for effective immune surveillance and the inflammatory response [176,177,294,295,296,297,298,299,300,301,302].

#### 6.2.3. Platelet and Endothelial Cell Adhesion Molecule 1 (PECAM1)

Platelet and endothelial cell adhesion molecule 1 (PECAM1), also known as CD31, is a 130 kDa protein [303,304,305]. PECAM1 is primarily expressed on endothelial cells and platelets, but its expression has also been observed in monocytes, dendritic cells, neutrophils, T cells, B cells, and NKT cells [303,305,306,307,308,309,310,311,312,313,314,315,316]. The interaction of PECAM1 with integrin αvβ3, VCAM1, and ICAM1 is crucial for mediating the adhesion of leukocytes to endothelial cells, facilitating cell migration during inflammation [317,318]. Additionally, PECAM-1 interplay with counter-receptors such as CD38 and CD177 suggests a role in the migration of CD38^+^ B cells and CD177^+^ neutrophils [319,320]. This complex network underscores the significant involvement of PECAM1 in immune cell trafficking and inflammation in Fabry disease. However, further studies are needed to confirm the expression of these adhesion molecules on platelets or other cell-derived macrovesicles present in peripheral blood from the mouse models and patients with Fabry disease

In Fabry disease, elevated levels of ICAM1, VCAM1, and PECAM1 have been observed in peripheral blood cells [77,78,79]. Additionally, findings have shown that leukocyte PECAM1 expression is significantly higher in both untreated and enzyme replacement therapy-treated patients with Fabry disease compared to controls, with no significant difference between the untreated and treated groups [11]. Another study indicates that excess Gb3 on exposed endothelial cells leads to the upregulation of adhesion molecules like ICAM1 and VCAM1 in Fabry disease [10]. In addition to the elevated peripheral blood levels of monocytes, macrophages, granulocytes, T cells, and B cells [11,78,80,81,82,83], bone marrow, endomyocardial, and kidney biopsies from patients with Fabry disease showed an increased infiltration of CD68^+^ CD163^+^ subsets of macrophages, CD3^+^T cells, NK cells, and CD19^+^ and CD138^+^ B cells [11,82,84,85,86].

Several studies have shown the crucial roles of the C3a–C3aR and C5a–C5aR1 axes in the alteration of the expression of VCAM1, ICAM1, and PECAM1, impacting vascular permeability and promoting excessive immune cell infiltration [62,63,64]. For instance, C3-deficient mice exhibit reduced brain edema, microglial cell activation, and neutrophil infiltration following intracerebral hemorrhage. Similarly, in models of meningitis and West Nile virus infection, deficiencies in complement components like C1q, C3, and C5aR1 result in fewer cerebrospinal fluid leukocytes and less brain damage [321,322,323].

Furthermore, C3a deficiency or blockade with C3a receptor antagonists (C3aRA) showed marked reduction in the endothelial expression of VCAM1 and leukocyte (e.g., CD8^+^ T cells) infiltration in models of cerebral inflammation [324]. A C3aRA-treated mouse model of middle cerebral artery occlusion showed lower expression of ICAM1 on endothelial cells, decreased neutrophil infiltration in the ischemic zone, and smaller stroke volumes compared to vehicle-treated controls, suggesting that targeting C3aR effectively modulates stroke-related injury in ischemia/reperfusion scenarios [325]. The stimulation of human umbilical vein endothelial cells with C5a revealed that it induces an upregulation of ICAM1 and VCAM1 [326]. An increased expression of ICAM1 induced by C5a has been observed in choroidal endothelial cells [327]. Additionally, anti-C5a treatment suppressed ICAM1 upregulation, neutrophil infiltration, lung vascular permeability, and injury in IgG immune complex-induced lung injury models [328].

Integrating these observations into this study on complement function in Fabry disease provides a broader context for understanding the specific roles of complement components, specifically the C3a and C5a mediated induction of ICAM1, VCAM1, and PECAM1, that could directly cause excess tissue recruitment of various immune cells, (e.g., macrophages, T cells, and B cells), driving inflammatory processes and promoting tissue damage in Fabry disease.

### 6.3. Cadherins

Cadherins are a family of adhesion molecules, i.e., CDH1 (E cadherin), CDH2 (N cadherin), CDH3 (P cadherin), CDH5 (VE cadherin), and CDH11 (Cad11), which mediate calcium-dependent or calcium-independent cell–cell adhesion, influencing the tissue recruitment of immune cells [329,330,331,332,333]. E cadherin is predominantly expressed in epithelial cells and involved in maintaining epithelial integrity. During inflammation, its downregulation can contribute to an altered tissue recruitment of immune cells [329]. N cadherin is expressed in neural tissues and some endothelial cells and contributes to neuronal development and tissue remodeling. Its role in immune cell recruitment is less direct but may influence the tissue environment. P cadherin is found in epithelial cells and some endothelial cells, P cadherin is involved in maintaining cell–cell adhesion and may impact immune cell migration and tissue repair processes. VE cadherin is an endothelial cell-specific cadherin that plays an important role in the control of vascular organization [334,335]. Cadherin11 expressed on mesenchymal tissues, placenta, brain, lung, heart, osteoblasts, and synovial fibroblasts [336,337,338,339,340].

It has been shown that Gb3-treated endothelial cell lines exhibit an upregulation of N cadherin but a decreased expression of E cadherin [10,341]. Furthermore, C3a-stimulated tubular epithelial cells showed a downregulation of E cadherin [342]. Brain endothelial cells in WT mice showed the positivity of VE cadherin^+^ whereas brain endothelial cells of C3aR^−/−^ mice showed the absence of VE cadherin positivity [324]. Similarly, C5a-stimulated human umbilical vein endothelial cells, hepatocellular carcinoma cells, and retinal pigment epithelial cells caused a downregulation of E cadherin and CDH11 [326,343,344].

Under normal physiological conditions, a healthy endothelial barrier effectively restricts uncontrolled immune cell movement, primarily due to the role of E cadherin, a crucial adhesion molecule that maintains cellular integrity and forms tight junctions between epithelial cells [345,346]. In Fabry disease, the C3a–C3aR signaling axis could disrupt this balance by driving the loss of E cadherin. When E cadherin levels drop, the endothelial cells lose their structural cohesion, becoming more vulnerable to leukocyte adhesion and infiltration. This breakdown of the endothelial barrier not only facilitates excessive leukocyte transmigration into surrounding tissues but also amplifies inflammation and accelerates tissue damage, worsening the pathological progression of Fabry disease.

### 6.4. Selectins

Selectins are a family of adhesion molecules involved in the initial stages of leukocyte adhesion and rolling on endothelial cells [347]. Selectins are divided into P, E, and L selectins, originally based on which cell types they were found in: platelets, endothelial cells, and leukocytes (however, P selectin is also expressed on endothelial cells) [329]. P selectin mediates the initial rolling of leukocytes on the endothelium through interactions with carbohydrate ligands, (e.g., P selectin glycoprotein ligand 1, PSGL1) facilitating their recruitment to sites of inflammation [348]. E selectin interacts with PSGL1 and Sialyl Lewis X (a carbohydrate structure that plays a vital role in cell-to-cell recognition processes), on leukocytes, promoting their rolling and adhesion [348]. L selectin is expressed on leukocytes, which bind glycosylated ligands, such as PSGL1, on endothelial cells, mediating the rolling of leukocytes on the blood vessel walls and facilitating their extravasation into the tissues [348,349].

Complement components are essential in the regulation of von Willebrand factor (vWF), a critical protein that facilitates platelet adhesion and plays a significant role in the coagulation cascade. vWF is not only vital for normal hemostasis but is also recognized as a marker for thrombotic cardiovascular diseases [350,351,352]. It has been shown that C5a can swiftly enhance the expression of P selectin and vWF on the surface of human umbilical vein endothelial cells, thereby influencing the pathophysiology of various conditions, including sickle cell disease [353]

Notably, elevated levels of C5a (Table 1) and P selectin have been detected in the peripheral blood of individuals with Fabry disease [77,78] as well as increased secretion of both vWF and P selectin in both in vitro and in vivo mouse models of Fabry disease [354]. These observations imply that the activation of the C5a–C5aR1 axis may lead to the activation of the P selectin–vWF pathway in endothelial cells, creating a hypercoagulable state that could exacerbate vascular complications associated with Fabry disease.

Such complications can manifest in thrombotic events, including coagulation defects and strokes, as evidenced by studies in both mouse models and affected patients [355,356,357,358,359]. However, to deepen our understanding of these processes, further functional studies are essential to validate these findings and elucidate the underlying mechanisms at play.

The C5a-mediated stimulation of human umbilical vein endothelial cells causes an upregulation of E selectin [326]. The systemic activation of C5a in rats, achieved via an intravenous infusion of cobra venom factor has been shown to induce lung injury that is P selectin dependent, and this upregulation was almost completely blocked by prior complement depletion or by the infusion of anti-rat C5a [360,361].

The increased P selectin on endothelial cells facilitates the early stages of leukocyte recruitment. This initial adhesion is critical for leukocytes to roll more slowly and adhere firmly to the endothelium, which is necessary for their eventual extravasation into tissues. In the context of Fabry disease, C3a–C3aR- and C5a–C5aR1-mediated upregulation of P selectin could enhance leukocyte adhesion and rolling on the endothelium, leading to an increased tissue recruitment of leukocytes. This process amplifies inflammation and contributes to the progression of tissue damage in Fabry disease.

## 7. Discussion

Complement activation is not merely a defensive mechanism; it is a double-edged sword that can inadvertently contribute to tissue inflammation in a range of visceral and central nervous system diseases [362,363,364]. The cascade of events initiated by complement activation is profound: the cleavage of C3 into its active fragments, C3a and C3b, sets in motion a series of reactions terminating in the formation of C5 convertase and the subsequent release of C5a [365,366,367,368,369,370]. This obscure interaction emphasizes the delicate balance within the immune system, where protective responses can sometimes exacerbate pathological conditions due to the excess tissue deposition of certain lipids, such as triglycerides and cholesterol [158,159,160,161,162,163,164,165,166].

In the context of Fabry disease, recent findings have revealed that enzyme replacement therapy lowers the Lyso-Gb3 but fails to fully mitigate complement activation. In fact, studies indicate that complement activation may intensify in these patients, suggesting the critical role of Gb3, rather than Lyso-Gb3, in driving this process [21].

Our previous investigations into Gaucher disease have further illustrated this dynamic, demonstrating a connection between glucosylceramide accumulation and local C5a production by immune cells such as macrophages and dendritic cells [167]. This raises an important question: could the excessive buildup of Gb3 in untreated Fabry patients similarly trigger the alternative pathway of complement activation? If so, this would lead to an increased local production of C3a and C5a, reinforcing the cycle of inflammation and contributing to the progression of the disease. This potential pathway emphasizes the need for a deeper understanding of how specific lipid accumulations cause complement activation, thereby leading to an increase in the local production of C3a and C5a in Fabry disease.

The immunological landscape of mice and human IgGs reveals critical distinctions that influence immune responses and therapeutic outcomes. Mouse IgGs (IgG1, IgG2a/c, IgG2b, and IgG3) and their corresponding receptors, such as FcγRI, FcγRIIb, FcγRIII, and FcγRIV, differ significantly from human IgGs (IgG1, IgG2, IgG3, and IgG4) and their receptors (including FcγRI, FcγRIIa, FcγRIIc, FcγRIIIa, FcγRIIb, and FcγRIIIb) [371,372]. Notably, while both mouse and human FcγRI exclusively bind monomeric IgG, other FcγRs in both species can interact with immune complexes (ICs) formed by IgG, emphasizing the complexity of these interactions. The crosslinking of IgG2a/c and IgG2b ICs to FcγRIII and FcγRIV in mice, alongside IgG1 ICs binding to activating FcγRs in humans, is particularly crucial for facilitating optimal complement activation [167,371,373,374,375,376,377,378].

In our previous findings of Gaucher disease, we observed a striking increase in serum levels of glucosylceramide-specific IgG2a/c and IgG2b autoantibodies, with IgG2a/c levels surpassing those of IgG2b. Correspondingly, elevated C5a levels were detected in the serum of affected mice. In human patients, we noted similar trends, with elevated levels of glucosylceramide-specific IgG1, IgG2, and IgG3 autoantibodies, and again, increased serum C5a, with IgG1 showing the highest levels. More critically, our investigations demonstrated that the formation of glucosylceramide-specific IgG-ICs leads to significant macrophage stimulation, both in vivo and ex vivo mouse models of Gaucher disease and in vitro in human cell models. This interaction catalyzed a massive production of C5a, suggesting the potential role of glucosylceramide-specific IgG-ICs in the induction of complement activation and C5a production in Gaucher disease [144,167,379].

In the context of Fabry disease, enzyme replacement therapies such as agalsidase α and β have been linked to the emergence of anti-drug IgG autoantibodies (Drug-IgG), including those of the IgG1 and IgG4 subclass [168,169,170]. The recognition of the drug by Drug-IgG leads to the formation of drug-specific IgG immune complexes (Drug-IgG ICs). These immune complexes activate the complement system via the classical pathway, resulting in an increased production of C3a and C5a in patients after treatment, as illustrated in Figure 2.

Leukocyte-mediated inflammation plays a pivotal role in the pathology of Fabry disease, with distinct contributions from various immune cells, including granulocytes, B and T lymphocytes, monocytes, and macrophages. Each of these cell types respond differently to the disease environment, influencing the overall inflammatory response and leading to vascular complications and tissue damage [11,13,380,381]. Granulocytes are often among the first responders to inflammation [382,383]. In Fabry disease, the activation of these immune cells can lead to the release of reactive oxygen species and proteolytic enzymes, contributing to endothelial damage and promoting vasculopathy. This can impair blood flow and exacerbate ischemic conditions in affected tissues. B and T lymphocytes also participate in the inflammatory response, with T cells potentially driving chronic inflammation through cytokine release. This can lead to a persistent inflammatory state, which further contributes to vascular dysfunction and tissue injury. B cells may produce autoantibodies that amplify this response, complicating the disease process [384,385]. Monocytes and macrophages play crucial roles in tissue remodeling and repair [386,387]. Macrophages can also contribute to further endothelial dysfunction through the secretion of cytokines and growth factors [388].

The recruitment of such leukocytes from the bloodstream to sites of inflammation is a finely orchestrated process involving a series of well-regulated interactions between cell adhesion molecules on both leukocytes and endothelial cells. This complex cascade encompasses rolling, firm adhesion, and transmigration, each stage governed by specific adhesion molecules.

Rolling is the initial step, mediated by selectins, which are membrane glycoproteins that bind to carbohydrate structures like PSGL1 on both leukocytes and endothelial cells [190]. This transient adhesion allows leukocytes to roll slowly along the endothelial surface, setting the stage for subsequent phases of adhesion. This rolling phase is decisive, as it prepares leukocytes for a stronger, more stable attachment [176,178,389,390]. As leukocytes roll, they transition to firm adhesion through the activation of integrins, which are initially in an inactive state. These integrins, including α4β1, αLβ2, CR3, αXβ2, CR4, and α4β7, bind with high affinity to their ligands on endothelial cells, such as ICAM1, ICAM2, ICAM3, VCAM1, and iC3b. This interaction significantly strengthens the adhesion, moving from transient rolling to firm attachment [293,298,391,392,393,394,395,396,397,398,399,400,401,402,403].

The firm adhesion phase is characterized by interactions between integrins and intercellular adhesion molecules. For example, α4β1 binds to VCAM1, and αLβ2 interacts with ICAM1, facilitating a robust connection between leukocytes and endothelial cells. This strong adhesion is critical for the subsequent step: transmigration [174,179,190,230,246,247]. To initiate transmigration, leukocytes form a transmigratory cup on the endothelial surface. This structure is created by the campaigning of LFA1, CR3, and VLA4, which colocalize with ICAM1 and VCAM1 on endothelial microvilli-like projections. This allows leukocytes to squeeze between endothelial cells and migrate into the tissue matrix [173,281,392,404,405,406,407,408].

In the context of Fabry disease, elevated levels of adhesion molecules such as VLA4, CR3, ICAM1, VCAM1, and PECAM1 in the bloodstream are closely linked to disease severity [11,77,78]. Moreover, studies observed an increase in blood lymphocyte counts, with notable elevations in specific T cytotoxic cell subsets (CCR4^+^CXCR3^+^ and CCR6^+^), as well as the presence of MHCII^+^ CD1d^+^ CD11b^+^ CD31^+^ monocytes in patients with Fabry disease [11,78,80,81]. The pathological impact of these immune processes extends far beyond the bloodstream, infiltrating vital organs such as bone marrow, heart, and kidneys. Biopsies from these tissues reveal significant immune cell infiltration, including macrophages, T cells, NK cells, and B cells [11,82,84,85,86]. Notably, many of these immune cells, such as monocytes, macrophages, and T and B cells express C3aR and C5aR1, highlighting their potential migration towards the tissue expressing the potent chemoattractants C3a and C5a that process the tissue damage in patients with Fabry disease [99,101,102,103,104,105,106,107,108,109,110,111,112,113,114,115,116,117,125,126,127,128,129,130,131,132,133,134].

The mechanisms by which the C3a–C3aR and C5a–C5aR1 axes drive excessive leukocyte recruitment in Fabry disease remain critical areas of research. Our findings indicate that C3a and C5a interact with their respective receptors, C3aR and C5aR1, on leukocytes (Figure 3a) and endothelial cells (Figure 3b). This interaction leads to the upregulation of various adhesion molecules. Specifically, on leukocytes, this is the increased expression of P selectin, VCAM 1, ICAM 1, and CR3 (Figure 3c), while on endothelial cells it is PSGL 1, VLA 4, VCAM 1, PECAM 1, and ICAM 1 (Figure 3d).

This sequential activation of adhesion molecules facilitates a multi-step process of leukocyte recruitment. Initially, leukocytes roll along the endothelial surface through P selectin–PSGL 1 interactions (Figure 3e). This rolling is followed by weak adhesion mediated by VLA 4 binding to VCAM 1 (Figure 3f). Subsequently, strong and firm adhesion occurs as ICAM-1 and VCAM-1 on leukocytes bind to PECAM-1 on endothelial cells (Figure 3g), along with CR3 interacting with ICAM 1 (Figure 3h). These coordinated interactions enable effective transmigration through the endothelial barrier, leading to the pronounced tissue infiltration characteristic of Fabry disease (Figure 3i).

The recognition of C3a and C5a by their respective receptors initiates a cascade of events that profoundly impact leukocyte and endothelial cell dynamics. These functions were observed in the pathogenesis of several of the inflammatory and lysosomal storage diseases. In autoimmune diseases such as rheumatoid arthritis and lupus, complement activation and the production of C3a and C5a contributes to chronic inflammation and tissue destruction [69,409,410]. In liver and lung injuries, C3a- and C5a-mediated effector function causes tissue damage [67,162,411,412,413,414,415,416]. In ischemia–reperfusion injury/stroke, complement activation and the action of C3a and C5a are implicated in tissue damage [70,417,418,419,420]. In renal diseases like glomerulonephritis, C3a and C5a activation leads to inflammation and kidney damage [65,421,422,423,424,425]. In cardiovascular diseases, C3a and C5a promotes the recruitment of inflammatory cells to the vascular wall, leading to hypertension and atherosclerosis [426,427,428]. In neurological diseases, C3a and C5a contribute to blood–brain barrier damage and leukocyte infiltration-mediated neuroinflammation and neuronal cell damage [63,324,325,429,430]. In situations like fibrosis and pain, C3a and C5a propagate the disease pathology [324,326,327,431]. In lysosomal storage diseases, like Gaucher and Niemann-Pick type C (NPC) diseases, C3a and C5a lead to tissue damage and the propagation of the disease [76,144,167,379,432,433,434].

The roles of C3a and C5a may be significant in the pathogenesis of Fabry disease. In affected individuals, there is a notable increase in C3a and C5a, an upregulation of adhesion molecules, and an excessive recruitment of various immune cells, including monocytes, macrophages, T cells, and B cells, as well as elevated levels of pro-inflammatory cytokines such as IL1α, IL1β, IL2, IL6, IL8, TNFα, and IL17, alongside endothelial dysfunction [10,78,98,99,100,101,435,436,437]. Moreover, in Fabry disease, infiltrated leukocytes may become activated in response to the effector functions of C3a and C5a. This activation leads to a shift toward a pro-inflammatory phenotype, perpetuating inflammation and exacerbating endothelial dysfunction in affected tissues. This ongoing inflammatory process can result in significant organ dysfunction, ultimately manifesting in a range of clinical symptoms, including renal failure, cardiovascular complications, gastrointestinal disturbances, angiokeratomas, strokes, and various neurological symptoms established in Fabry disease [9,10,11,12,13,14,15,16,17,26,27,28,29,31,32,33,34,35,36,37,38,39,40,438]

Endothelial dysfunction is a critical concern in Fabry disease, significantly contributing to the disease pathology and its clinical manifestations. In affected individuals, the endothelial cells lining blood vessels become impaired, leading to increased vascular permeability, inflammation, and impaired vasodilation. This dysfunction is often driven by a combination of factors, including the excessive activation of immune cells and pro-inflammatory cytokine production. As a result, patients experience complications such as renal failure, cardiovascular issues, and ischemic events [439,440,441,442,443]. The exact mechanism of endothelial dysfunction in Fabry disease remains unclear.

Pollmann et al. investigated the endothelial glycocalyx in Fabry patients and healthy controls undergoing enzyme replacement and substrate reduction therapies by measuring arterial stiffness and endothelial function. In vivo results indicated that enzyme replacement therapy or substrate reduction therapy improved glycocalyx thickness, red blood cell count, and small vessel function. In vitro, α-Gal A-deficient endothelial cells showed increased levels of nuclear factor kappa B (NF-kB) signaling, alongside a reduced glycocalyx and enhanced monocyte adhesion. Wild-type cells exposed to pathological Gb3 displayed similar issues. Treatments with recombinant α-Gal A, heparin, anti-inflammatory, and antioxidant agents improved glycocalyx structure and endothelial function in these cells [22]. Additionally, Vahldieck et al. identified that C5a–C5aR1 signaling, i.e., RhoA → Rho associated protein kinase (ROCK) activation induces structural and mechanical changes in the endothelial glycocalyx, further exacerbating endothelial dysfunction [444].

The activation of inflammatory pathways, such as the C3a–C3aR and C5a–C5aR1 pathways, not only contributes to immune cell recruitment but also creates an environment that influences antigen processing and presentation [445,446,447]. In this context, two primary pathways for antigen presentation have been elucidated in the immune system: the presentation of endogenous antigens via major histocompatibility complex class I (MHC I) and the processing of exogenous antigens, such as those from intracellular pathogens, on MHC class II [448,449]. 

Both MHC I and MHC II are predominantly expressed by antigen-presenting cells (APCs) like monocytes, which play a fundamental role in delivering antigens to CD4^+^ and CD8^+^ T cells [450,451,452,453]. Notably, MHC II-positive monocytes have been shown to effectively present intravascular antigens to CD4^+^ T cells, and cause the production of IFNγ, TNFα, IL1α, IL1β, IL6, and IL17 [449,452,454,455,456,457,458,459].

Patients with Fabry disease also demonstrate an increase in T cell subsets and increased circulatory levels of pro-inflammatory cytokines, such as TNFα [12,77,79,82,460,461,462], IFNγ [82], IL1α [460], IL1β [12,77,460], IL6 [21,79,461,462], and IL17 [21]. This suggests a shift toward a pro-inflammatory milieu, which likely contributes to the pathogenesis of the disease. In addition, circulating monocytes in Fabry disease show upregulation of MHC II expression, alongside significantly elevated levels of soluble ICAM 1 and VCAM1 [11,78]. These findings raise important questions: Could the increased levels of soluble VCAM 1 and ICAM 1 act as decoys, diverting monocytes and T lymphocytes away from endothelial cell membranes? Moreover, do these elevated soluble markers indicate ongoing endothelial cell damage in Fabry disease? Exploring these questions could provide deeper insights into the mechanisms driving immune dysregulation in Fabry disease, underscoring the need for further research to confirm these hypotheses.

In parallel, the C3a and C5a complement proteins play a pivotal role in modulating immune responses by enhancing the function of antigen-presenting cells, (e.g., monocytes, macrophages, and dendritic cells) and T cells. These complement fragments upregulate key stimulatory and co-stimulatory molecules, including MHC II, CD40, CD80, CD86, CD40L, and CD69 [463,464,465,466]. We have shown in our previous studies that such activation not only boosts the overall immune response but also optimizes antigen processing and presentation, thereby amplifying the effectiveness of the innate and adaptive immune responses [144,167,467,468,469].

Together, these findings implicate the C3a–C3aR and C5a–C5aR1 pathways as key modulators in Fabry disease, driving the upregulation of MHC II on circulating monocytes. These MHC II-expressing monocytes, in turn, process circulating soluble ICAM 1 and VCAM 1, presenting them to CD4^+^ T cells. This process activates these immune cells and induces the release of additional pro-inflammatory cytokines, i.e., IL1α, IL1β, TNFα, IFNγ, and IL17, which may exacerbate endothelial dysfunction and contribute to the degradation of the endothelial glycocalyx. This cascade of immune events is central to the vascular abnormalities observed in Fabry disease, underscoring the crucial relationship between immune cell activation and disease pathology (Figure 4a–d). 

Our study discloses a compelling mechanism through which Gb3-induced activation of the C3a–C3aR and C5a–C5aR1 pathways ignites a cascade of immune responses that significantly contributes to the progression of Fabry disease. This complex cascade not only involves the upregulation of adhesion molecules but also facilitates increased leukocyte recruitment, creating a perfect storm of inflammation and tissue damage. Gaining a deeper understanding of these mechanisms could illuminate potential therapeutic targets, particularly within the C3–C3a–C5a signaling pathways and their receptor interactions. This insight may lead to strategies for abating excessive immune cell infiltration, ultimately improving disease outcomes.

The question of why autonomic and sensory neuropathies, along with gastrointestinal symptoms, often manifest in childhood in classical male Fabry disease, while clinically significant renal and cardiac damage tends to be delayed for several decades, invites a thoughtful exploration of underlying mechanisms. One potential explanation lies in the role of Gb3-induced complement activation. The accumulation of Gb3 in tissues can trigger the complement cascade, leading to the production of pro-inflammatory mediators such as C3a and C5a. These complement components may play a crucial role in propagating the early autonomic and sensory neuropathies and gastrointestinal symptoms observed in childhood. The nervous system, particularly the autonomic pathways, appears to be particularly sensitive to these inflammatory processes, which may explain the early onset of these symptoms [470,471].

In contrast, the development of renal and cardiac damage may be influenced by a different set of factors. Prolonged treatment with enzyme replacement therapy can result in the formation of Drug- IgG ICs. The formation of such Drug- IgG ICs can activate the classical pathway of the complement system, further leading to the generation of C3a and C5a, along with their downstream effector functions. This process may contribute to the gradual progression of renal and cardiac damage over decades, as the response of the immune system evolves in conjunction with the ongoing disease pathology.

Thus, the distinct timelines of symptom manifestation in Fabry disease can be understood as a complex back-and-forth between early complement-mediated inflammation affecting the nervous system and later immune responses related to enzyme replacement therapy impacting renal and cardiac health. This underlines the need for ongoing monitoring and altered therapeutic approaches that consider these differing mechanisms throughout the lifespan of affected patients. Emerging therapies that target the complement system, such as pegcetacoplan for C3, eculizumab, an anti-C5 monoclonal antibody, and Avacopan, a specific C5aR1 inhibitor, have shown remarkable clinical efficacy in a range of conditions, including paroxysmal nocturnal hemoglobinuria, hemolytic uremic syndrome, anti-neutrophil cytoplasmic antibody-associated vasculitis, refractory generalized myasthenia gravis, and neuromyelitis optica spectrum disorder [472].

Coppola et al. presented a compelling clinical case of a patient diagnosed with both atypical hemolytic-uremic syndrome and Fabry disease, illustrating the complexities involved in managing these interconnected conditions. Their findings reported that the timely administration of eculizumab has demonstrated significant efficacy in improving outcomes related to thrombotic microangiopathy and in protecting renal function in a patient diagnosed with both atypical hemolytic-uremic syndrome and Fabry disease [473]. This success suggests a promising opportunity to repurpose eculizumab, Avacopan, and similar treatments for complications arising from Fabry disease.

## Figures and Tables

**Figure 1 ijms-25-12252-f001:**
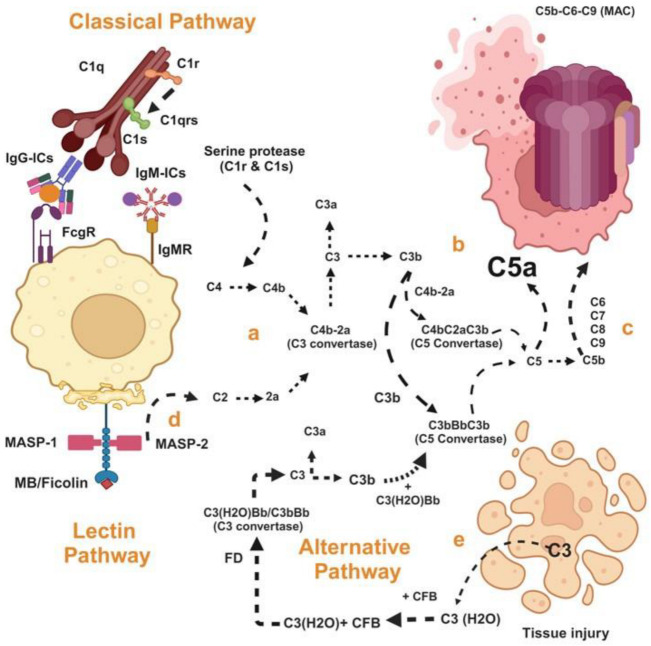
Complement pathways overview. This figure illustrates the three pathways of complement activation: the classical, lectin, and alternative pathways, each terminating in the formation of the complement 5a and membrane attack complex (MAC) and subsequent cell lysis. (**a**) The classical pathway is initiated when the C1 complex, composed of C1q, C1r, and C1s, binds to the Fc region of antibodies (IgM or IgG) that are attached to antigens or released from damaged cells. This binding activates the serine proteases C1r and C1s, leading to the cleavage of C4 and C2, resulting in the formation of C4b2a, a C3 convertase. (**b**) C4b2a cleaves C3 into C3a and C3b. C3b then binds to C4b2a to create C5 convertase (C4b2aC3b), which cleaves C5 into C5a and C5b. (**c**) C5b initiates the assembly of the MAC, comprising C6, C7, C8, and multiple C9 molecules, forming transmembrane channels that lead to cell lysis in targeted microorganisms or damaged host cells. (**d**) The lectin pathway is activated when mannose-binding lectin (MBL) or ficolins bind to carbohydrate patterns on pathogens or altered host cell surfaces. This triggers the activation of MBL-associated serine proteases (MASPs), which cleave C4 and C2, leading to the formation of C4b2a and subsequently the cleavage of C3 into C3a and C3b. The process mirrors that of the classical pathway, ultimately forming C5 convertase and the MAC. (**e**) The alternative pathway operates at a basal level and amplifies upon encountering pathogens or tissue injuries. It begins with the spontaneous hydrolysis of C3, forming C3 (H_2_O) that binds factor B, followed by cleavage by factor D to yield C3 (H_2_O)Bb, serving as a C3 convertase. This pathway also leads to the generation of C5 convertase and the assembly of the MAC. In the entire figure, solid, dashed, and bidirectional arrows are employed to illustrate the various components of the complement system and to show up the interconnected nature of the complement pathways, including their activation and amplification.

**Figure 2 ijms-25-12252-f002:**
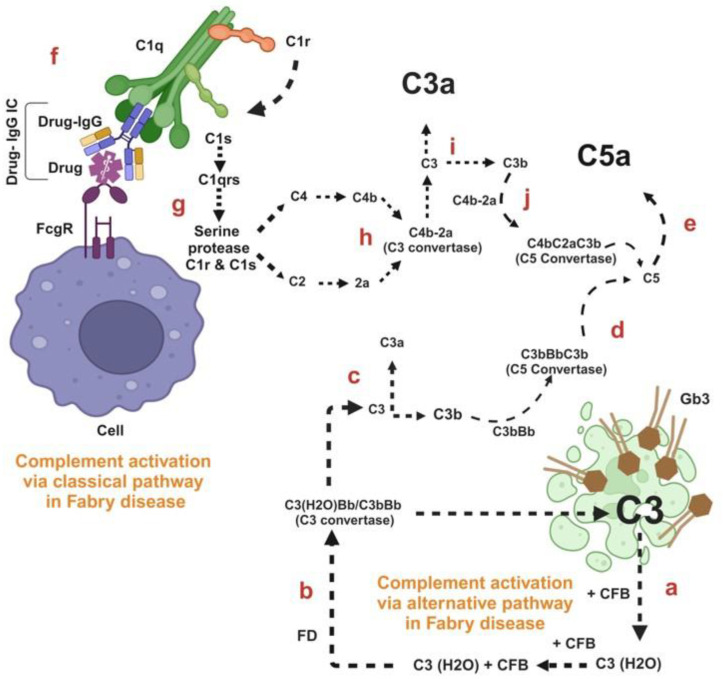
Mechanisms of complement activation in Fabry disease. (**a**) Excess accumulation of Gb3 may lead to spontaneous hydrolysis of C3 via the alternative complement activation pathway, resulting in the formation of C3 (H_2_O). (**b**). The binding of C3 (H_2_O) to complement factor B (CFB) initiates its cleavage by factor D (FD), producing the C3 convertase complex, C3 (H_2_O)Bb. This enzyme complex cleaves additional C3 molecules into C3a and C3b. (**c**). The generated C3b can further bind to CFB, which is then cleaved by FD to form the potent C3 convertase, C3bBb. (**d**) The activity of C3bBb facilitates the downstream cleavage of C3 into C3a and C3b. (**e**). Additionally, the interaction of C3bBb with C3b results in the formation of the C5 convertase, C3bBbC3b, leading to the cleavage of C5 into C5a within the context of Fabry disease. (**f**). Post-enzyme replacement therapy may trigger the production of drug-specific IgG antibodies (Drug-IgG), which bind to the therapeutic agents, agalsidase α or agalsidase β, forming drug-specific IgG immune complexes (Drug-IgG ICs). (**g**) The subsequent interaction of C1q with these Drug-IgG ICs activates serine proteases C1r and C1s. (**h**) This activation initiates the cleavage of C4 and C2, ultimately leading to the formation of the C3 convertase (C4b2a) through the classical complement pathway. (**i**) The resultant C3 convertase (C4b2a) facilitates the cleavage of C3 into C3a and C3b. (**j**) The interaction between the C3 convertase and C3b further results in the formation of the C5 convertase, which cleaves C5 into C5a, particularly in the context of Fabry disease. In the figure, dashed arrows are used to represent the components of the complement system involved in complement activation via the classical and alternative pathway in Fabry disease.

**Figure 3 ijms-25-12252-f003:**
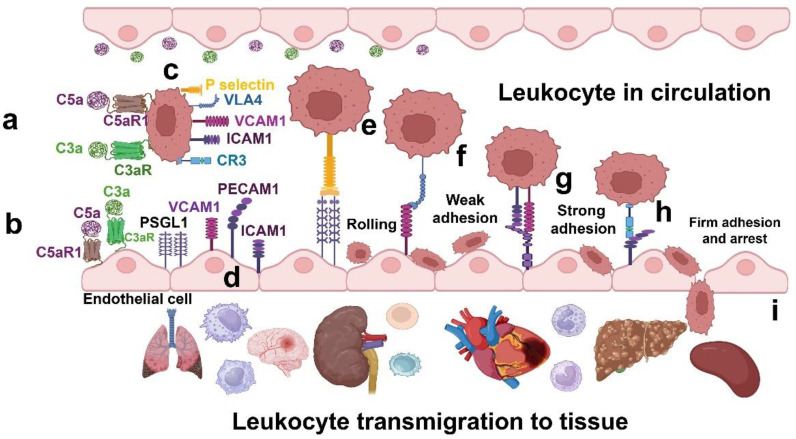
The C3a–C3aR and C5a–C5aR1 axes mediate excessive leukocyte recruitment in Fabry disease. (**a**,**b**) In Fabry disease, C3a and C5a bind to their respective receptors C3aR for C3a and C5aR1 for C5a on both leukocytes and endothelial cells. (**c**) Such C3a–C3aR and C5a–C5aR1 axes activation triggers a cascade of cellular responses. This interaction upregulates key adhesion molecules including P selectin, Very Late Antigen 4 (VLA 4), vascular cell adhesion molecule 1 (VCAM1), intercellular adhesion molecule 1 (ICAM1), and complement receptor 3 (CR3) on leukocytes. (**d**) On endothelial cells, the same receptors induce the expression of P selectin-glycoprotein ligand 1 (PSGL1), VCAM1, platelet and endothelial cell adhesion molecule 1 (PECAM1), and ICAM1. (**e**) The sequential activation of these adhesion molecules facilitates the progression of leukocyte recruitment. Initially, leukocytes loosely roll along the endothelial surface via P selectin–PSGL1 interactions. (**f**–**h**) These rolling transitions from weaker to stronger, more stable adhesion through VLA4–VCAM1 interactions, followed by a series of firm adhesion events mediated by ICAM1-VCAM1, PECAM1, and CR3–ICAM1 interactions. (**i**) These interactions conclude in the transmigration of leukocytes through the endothelial barrier, exacerbating tissue damage in Fabry disease.

**Figure 4 ijms-25-12252-f004:**
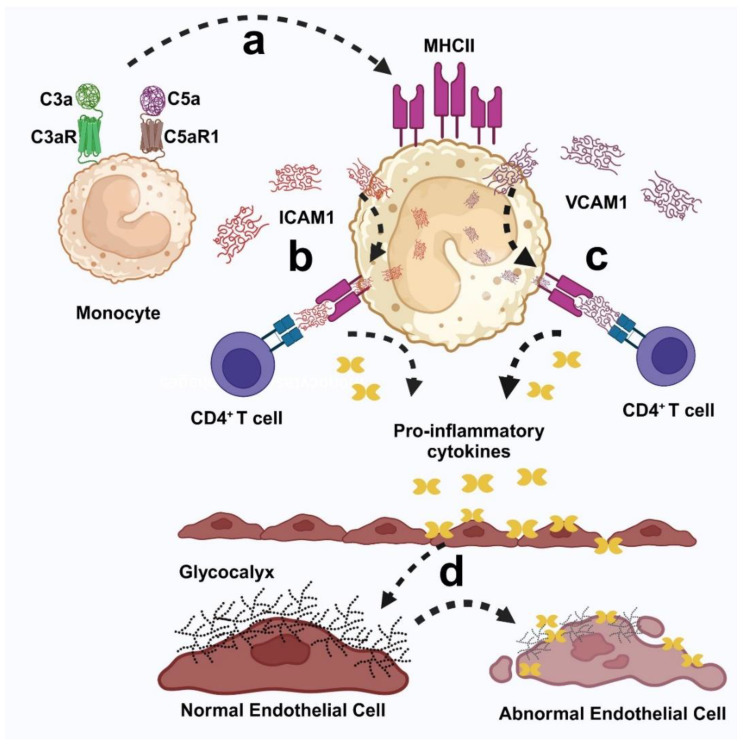
Immunological interactions and inflammatory pathways lead to vascular abnormalities in Fabry disease. (**a**) Circulating C3a and C5a engage their respective receptors, C3aR and C5aR, on monocytes, triggering the upregulation of the major histocompatibility complex class II (MHCII). (**b**) MHCII-expressing monocytes then process and present circulatory soluble intercellular adhesion molecule 1 (ICAM1) to CD4^+^ T cells, facilitating their activation and subsequent production of pro-inflammatory cytokines. (**c**) Simultaneously, MHC II-positive monocytes process and present circulatory soluble vascular cell adhesion molecule 1 (VCAM1)) to CD4^+^ T cells, further promoting cellular activation and enhancing pro-inflammatory cytokine production. (**d**) The cumulative effects of these immunological interactions and the resulting pro-inflammatory cytokines inflict damage on the glycocalyx, the protective layer of endothelial cells, leading to endothelial injury and contributing to the vascular abnormalities characteristic of Fabry disease.

**Table 1 ijms-25-12252-t001:** Complement activation in Fabry disease.

Involvement of Complement Components	Mouse Model of FD (Gla^−/−^)		Patients with FD	
	Source	References	Source	References
C1qc ^hi^			Plasma ^(P)^	[58]
C3 ^hi^	Renal tissue	[58]	Sera ^(P)^, mesangium ^(P)^, glomerular basement membrane ^(P)^, hilar arteriole ^(P)^, and brain ^(P)^	[15,59]
iC3b ^hi^	Plasma	[58]	Plasma ^(P)^	[58]
C4/C4b ^hi^			Plasma ^(P)^, sera ^(P)^	[15,58]
CFB precursors (C3/C5 convertase) ^hi^			Sera ^(P)^	[15]
C3a ^hi^			Sera ^(P)^	[21]
C5a ^hi^			Sera ^(P)^	[21]

FD (Fabry disease), C1qc (complement 1q subcomponent c), C3 (complement component 3), iC3b (inactivated form of complement component 3b), C4/C4b (complement components C4 and C4b), CFB (complement factor B), C3a (complement 3a), and C5a (complement 5a), P (protein expression), hi (higher levels).

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
