# Peer review of "Complement System and Adhesion Molecule Skirmishes in Fabry Disease: Insights into Pathogenesis and Disease Mechanisms"

_ijms, 2024, doi:10.3390/ijms252212252_

Round 1
Reviewer 1 Report
Comments and Suggestions for Authors
The author has written a comprehensive review of current knowledge of the role of complement activation in contributing to the chronic inflammatory state that appears to be a significant component of the pathophysiology of Fabry disease. My major suggestion for improvement relates to the discussion section. As written, it reiterates too much of the detail that was already included in the body of the text. I would delete much of that in favor of a briefer summary of the findings and use the rest of the discussion to put this work into a broader context.
Granted that complement-driven leukocyte mediated inflammation (which is likely different for neutrophils, B/T lymphocytes, monocytes, macrophages) is an important element in FD pathophysiology, there are other questions that need to be answered. For example, how does intracellular accumulation of GB3 and other secondary glycosphingolipids lead to an excess of C3a and C5a (step 1 in the figure)? Where do the assortment of cytokines that have been found to be increased in FD fit into the model? Aside from inflammation, does endothelial dysfunction also result in a coagulopathy via vWF that can exacerbate damage at the tissue and organ levels? (Kang JJ, Kaissarian NM, Desch KC, Kelly RJ, Shu L, Bodary PF, Shayman JA. α-galactosidase A deficiency promotes von Willebrand factor secretion in models of Fabry disease. Kidney Int. 2019 Jan;95(1):149-159. doi: 10.1016/j.kint.2018.08.033.) Aside from recruited leukocytes, do tissue-resident macrophages participate in the inflammatory process and directly impact tissue cells such as cardiomyocytes.
I have a number of other comments and suggestions that I highlighted in comments in the text itself that I will upload.
In a couple of places, you make reference to "in our study." Were you referring to this review article or to work that you have previously published?

I noticed several typos or grammatical errors that I highlighted in the uploaded file.
Author Response
I sincerely appreciate the reviewer’s positive feedback and insightful recommendations, which have been invaluable in refining the manuscript's focus and clarity.
I want to address all the interests raised pointwise.
Reviewer 1
The author has written a comprehensive review of current knowledge of the role of complement activation in contributing to the chronic inflammatory state that appears to be a significant component of the pathophysiology of Fabry disease. My major suggestion for improvement relates to the discussion section. As written, it reiterates too much of the detail that was already included in the body of the text. I would delete much of that in favor of a briefer summary of the findings and use the rest of the discussion to put this work into a broader context.
Granted that complement-driven leukocyte-mediated inflammation (which is likely different for neutrophils, B/T lymphocytes, monocytes, and macrophages) is an important element in FD pathophysiology, other questions need to be answered. For example, how does the intracellular accumulation of GB3 and other secondary glycosphingolipids lead to an excess of C3a and C5a (step 1 in the figure)? Where does the assortment of cytokines that are increased in FD fit into the model? Aside from inflammation, does endothelial dysfunction also result in coagulopathy via vWF that can exacerbate damage at the tissue and organ levels? (Kang JJ, Kaissarian NM, Desch KC, Kelly RJ, Shu L, Bodary PF, Shayman JA. α-galactosidase A deficiency promotes von Willebrand factor secretion in models of Fabry disease. Kidney Int. 2019 Jan;95(1):149-159. doi: 10.1016/j.kint.2018.08.033.) Aside from recruited leukocytes, do tissue-resident macrophages participate in the inflammatory process and directly impact tissue cells such as cardiomyocytes?
I have several other comments and suggestions that I highlighted in comments in the text itself that I will upload.
Comments cited in PDF MS:
Line 74: particularly in renal failure, cardiomegaly, stroke, neurodegenerative disorders, and lysosomal storage diseases. Is this a general statement or specific for Fabry disease
Response: This statement is general and not specific to Fabry disease. The correction has been made in the revised manuscript.
Line 74-76: The question raised whether elevated circulatory levels of leukocytes exist in Fabry patients in sterile conditions.
Response:
Elevated blood lymphocyte counts, along with increased levels of specific T cytotoxic cell subsets (CCR4+CXCR3+ and CCR6+) have been observed in patients with Fabry disease. Additionally, the presence of MHCII+ CD1d+ CD11b+ CD31+ monocytes has been noted. Importantly, the referenced studies do not indicate any correlation with infections (PMID: 19259782, PMID: 38623626, PMID: 30972193, and PMID: 10665494). We have incorporated these findings into the Discussion section.
Line 114 is shown in the PDF copy
Response: Suggested corrections have been completed (C5aR1 expression has been identified in a diverse array of immune cells, including monocytes, macrophages, dendritic cells, neutrophils, mast cells, T cells, and B cells, as well as in non-immune cells such as endothelial cells, mesangial cells, hepatocytes, keratinocytes, and neuronal cells. This expression is also observed across various tissues, including the liver, spleen, lungs, and brain126-135)
Line 118 shown in PDF copy
Response: Typos have been corrected (C5aR2 shares 58% sequence homology with C5aR1 and has a molecular weight of approximately 37 kDa126).
Line 125 is shown in the PDF copy
Response: Il12 has been corrected with IL12.
Line 150: One important question is whether complement activation and the processes that precede and are downstream are sporadically or continually active in Fabry patients, whether are they directly the consequence of GL3 and/or lyso-GL3 tissue content, and whether accumulation in specific tissues is more critical than in others to the overall systemic effects. Does the complement activation process accelerate with age and does tissue damage itself fuel the process?
Response: We appreciate the reviewer’s comments regarding the dynamics of complement activation in Fabry disease and its relation to aging. We have revised the paper by including the following additions.
(Aging is a well-established risk factor for various conditions, including age-related macular degeneration, metabolic disorders, and neurodegenerative diseases, all of which are influenced by complement activation. In this context, Fabry disease is particularly notable due to its association with reduced life expectancy and indications of premature aging. This raises critical questions about the relationship between aging, the production of complement components C3a and C5a, and disease progression.
Interestingly, a study by Laffer et al. revealed that levels of C3a and C5a in patients with Fabry disease do not correlate with age. This finding suggests that the dysregulation of these complement components may persist across different age groups. Therefore, further investigation into how complement activation contributes to disease mechanisms in the context of aging and Fabry disease is warranted.
Importantly, C3a concentrations above 5,000 ng/ml were observed in all treatment-naive patients with Fabry disease, and this elevated level persisted even after enzyme replacement therapy, particularly in individuals with a marked reduction in lyso-Gb3 and positive drug-specific IgG antibodies. These findings collectively suggest that complement activation in Fabry disease is dependent on Gb3, but not necessarily on lyso-Gb3, to produce C3a and C5a).
Line 184: What is the ligand for podocyte VL3A?
Response: The ligand for podocyte α3β1 integrin, specifically the VL3A epitope, is laminin-521. This interaction plays a crucial role in maintaining the structural integrity and function of podocytes in the glomerular basement membrane. Laminin-521 is essential for cell adhesion and signaling, and its binding to α3β1 integrin facilitates the connection between podocytes and the extracellular matrix, thereby supporting podocyte health and function. All this information have been added
Line 196: Can circulatory VCAM1 act as a decoy that diverts monocytes and lymphocytes from binding to endothelial cell membranes? Are elevated levels of circulatory VCAM1 and ICAM1 (see below) in Fabry disease a function of endothelial cell damage?
Response: There are great chances that elevated ICAM1 and VCAM1 in circulation may indeed be recognized by MHC II-positive monocytes, facilitating allo-monocyte-mediated processing and presentation to T cells. This interaction can lead to the activation of both monocytes and T cells, resulting in the production of certain cytokines such as IL1α, IL1β, IL6, TNFα, IFNγ, and IL17, which are all present in the Fabry patient's circulation and can damage endothelial cells and contribute to vascular abnormalities affecting multiple organs in Fabry disease.
We have added the following section and additional related references to our manuscript.
“Endothelial dysfunction is a critical concern in Fabry disease, significantly contributing to the disease pathology and its clinical manifestations. In affected individuals, the endothelial cells lining blood vessels become impaired, leading to increased vascular permeability, inflammation, and impaired vasodilation. This dysfunction is often driven by a combination of factors, including the excessive activation of immune cells and pro-inflammatory cytokines production. As a result, patients experience complications such as renal failure, cardiovascular issues, and ischemic events448-452. The exact mechanism of endothelial dysfunction in Fabry disease remains unclear.
Pollmann et al. investigated the endothelial glycocalyx in Fabry patients and healthy controls undergoing enzyme replacement and substrate reduction therapies by measuring arterial stiffness and endothelial function. In vivo, results indicated that enzyme replacement therapy or substrate reduction therapy improved glycocalyx thickness, red blood cell count, and small vessel function. In vitro, α-Gal A-deficient endothelial cells showed increased levels of NF-kB signaling, alongside a reduced glycocalyx and enhanced monocyte adhesion. Wild-type cells exposed to pathological Gb3 displayed similar issues. Treatments with recombinant alpha-galactosidase A, heparin, anti-inflammatory, and antioxidant agents improved glycocalyx structure and endothelial function in these cells22. Additionally, Vahldieck et al. identified that C5a- C5aR1 signaling, (e.g., RhoA→ ROCK) activation induces structural and mechanical changes in the endothelial glycocalyx, further exacerbating endothelial dysfunction453.
The C3a-C3aR and C5a-C5aR1 pathways are crucial in mediating inflammatory responses through the activation of NF-κB and RhoA signaling 454-456. This activation leads to the recruitment and activation of monocytes and other immune cells, resulting in the release of pro-inflammatory cytokines457,458
Two primary pathways for antigen presentation have been elucidated in the immune system: the presentation of endogenous antigens via major histocompatibility complex class I (MHC I) molecules and the processing of exogenous antigens, such as those from intracellular pathogens, on MHC class II molecules459,460. Both MHC I and MHC II are predominantly expressed by anti-gen-presenting cells (APCs) like monocytes, which play a fundamental role in delivering antigens to CD4+ and CD8+ T cells 461-464. Notably, MHC II positive monocytes have been shown to effectively present intravascular antigens to CD4+ T cells and cause the production of IFNγ, TNFα, IL1α,465 IL1β, IL6, and IL17460,463,466-470
In the context of Fabry disease, compelling evidence indicates that circulating monocytes exhibit upregulated expression of MHC II, alongside significantly elevated circulatory levels of ICAM 1 and VCAM111,79. Patients with Fabry disease also demonstrate an increase in T cell subsets and increased circulatory levels of pro-inflammatory cytokines, such as TNFα12,78,80,83,471-473, IFNγ83, IL1α471, IL1β12,78,471, IL621,80,472,473, and IL1721 suggesting a shift toward a pro-inflammatory environment within the context of the disease”.
Addition of new references.
- Pollmann S, Scharnetzki D, Manikowski D, Lenders M, Brand E. Endothelial Dysfunction in Fabry Disease Is Related to Glycocalyx Degradation. Front Immunol 2021; 12: 789142.
- Hwang A-R, Park S, Woo C-H. Lyso-globotriaosylsphingosine induces endothelial dysfunction via autophagy-dependent regulation of necroptosis. The Korean Journal of Physiology & Pharmacology: Official Journal of the Korean Physiological Society and the Korean Society of Pharmacology 2023; 27(3): 231-40.
- Park JL, Whitesall SE, D’alecy LG, Shu L, Shayman JA. VASCULAR DYSFUNCTION IN THE α‐GALACTOSIDASE A‐KNOCKOUT MOUSE IS AN ENDOTHELIAL CELL‐, PLASMA MEMBRANE‐BASED DEFECT. Clinical and Experimental Pharmacology and Physiology 2008; 35(10): 1156-63.
- Kang JJ, Shu L, Park JL, Shayman JA, Bodary PF. Endothelial nitric oxide synthase uncoupling and microvascular dysfunction in the mesentery of mice deficient in α-galactosidase A. American Journal of Physiology-Gastrointestinal and Liver Physiology 2014; 306(2): G140-G6.
- Namdar M, Gebhard C, Studiger R, et al. Globotriaosylsphingosine accumulation and not alpha-galactosidase-A deficiency causes endothelial dysfunction in Fabry disease. PloS one 2012; 7(4): e36373.
- Stamerra CA, Del Pinto R, di Giosia P, Ferri C, Sahebkar A. Anderson–Fabry Disease: From Endothelial Dysfunction to Emerging Therapies. Advances in Pharmacological and Pharmaceutical Sciences 2021; 2021(1): 5548445.
- Vahldieck C, Löning S, Hamacher C, et al. Dysregulated complement activation during acute myocardial infarction leads to endothelial glycocalyx degradation and endothelial dysfunction via the C5a:C5a-Receptor1 axis. Frontiers in Immunology 2024; 15.
Line 214: Has the distribution of classical v non-classical monocytes been investigated in Fabry disease? The non-classical monocytes which appear to be very important for endothelial repair express higher levels of CD18 than the classical monocytes. Also relevant to the next paragraph. See Narasimhan PB, Marcovecchio P, Hamers AAJ, Hedrick CC. Nonclassical Monocytes in Health and Disease. Annu Rev Immunol. 2019 Apr 26;37:439-456. doi: 10.1146/annurev-immunol-042617-053119. PMID: 31026415.
Response: Currently, the clear characterization of classical, intermediate, and non-classical monocyte populations has not been thoroughly investigated in the context of Fabry disease. However, based on previous findings, it appears that Fabry patients exhibit both classical CD14+ and intermediate MHC II+ monocytes. The role of non-classical monocytes, which are important for endothelial repair and express higher levels of CD18 than classical monocytes, merits further exploration in Fabry disease.
Line 260: typo? endothelial cells?
Response: Typo has been corrected to endothelial cells
Line 304:
Response: Polymorph nuclear cells have been corrected with neutrophils
Line 344: Typess
Response: Typess has been corrected to Types
Line314: Do you mean "from the bloodstream. to.."
Response: This paragraph has been corrected.
Line 351: grammar? been observed
Response: Grammer corrections have been completed.
Line 357: Is it certain that these membrane-expressed adhesion molecules are not associated with platelet or other cell-derived microvesicles when detected in the peripheral blood?
Response: Certain adhesion molecules may be expressed on platelets or other cell-derived microvesicles when detected in peripheral blood.
Line 360: On review of reference 11, I did not see supporting evidence for this statement. regarding FD pathology. and PECAM leukocyte expression. that was significantly higher in both untreated and ERT-treated FD patients compared with controls but not t different between non-treated and treated patients. No correlation studies were done based on FD Mainz severity scores.
Response: Thanks for catching the error and providing precious feedback. Suggested corrections have been completed
Line 366: omit the highlighted words, The point is that the tissue infiltration by these various cells was likely dependent on the expression of the adhesion molecules. as mentioned in the next sentence.
Response: Corrected
Line 383: C5a-induced (grammar correction)
Response: Corrected with the addition of -Increased expression of ICAM1 induced by C5a has been observed in choroidal endothelial cells
Line 390: instead of "involve in," cause would be better.
Response: This has been corrected to use “cause” instead of “involve in”
Line 416: Incomplete sentence
Response: The incomplete sentence has been corrected.
Line 428: This can be deleted.
Response: The suggested deletion has been made.
Line 449: Will there be discussion of FD and coagulation (eg vWF and P-selectin).
Response: Thanks for catalyzing the inclusion of this very interesting aspect, which could be a new area of research. I have included the C5a-C5aR1 role in vWF-mediated activation of the coagulation cascades in Fabry disease.
Line 487: Immune response or inflammatory response? They are not identical. Immune complexes have been reported in some FD biopsies. (Lerner YV, Tsoy LV, Grishina AN, Varshavsky VA. Morfologicheskaya kharakteristika izmenenii pochek pri bolezni Fabri [Morphological characteristics of renal
Response: This entire paragraph has been revised for clarity regarding the distinction between immune response and inflammatory response.
Line 458: Add: and their receptors.
Response: Corrected
Line 498: What mechanisms have been either proposed or demonstrated to explain how GB3 lysosomal storage leads to complement over production. What are the intermediate steps? What about the role of cytokines?
Response: The precise mechanisms of complement activation and the production of C3a and C5a have been included. Please refer to the new Figures 1 and 2 for details.
Line 517: Could you speculate as to why there have not been trials of available complement inhibitors such as eculizumab in FD patients. I found none on clinicaltrials.gov. and did not find a reference to looking at eculizumab in Fabry mice. There is a report of eculizumab use in a patient who had both FD as well as HUS. However, the paper is in Italian.
Coppola S, Cuomo V, Riccio CG, d'Apice L, de Simone W, Capasso G. [The unusual couple: a clinical case of coexistence between aHUS and Fabry's disease]. G Ital Nefrol. 2019 Feb;36(1):2019-vol1. Italian. PMID: 30758152.
Response: The absence of clinical trials involving complement inhibitors, such as eculizumab, in patients with Fabry disease is indeed intriguing and deserves further investigation. anti-C5 monoclonal antibodies, eculizumab have been approved by the FDA, with several others currently in various stages of clinical trials (PMID: 37979593). Given the availability of these pharmaceutical agents, it would be relatively straightforward to evaluate their effects on the inflammatory phenotype of Fabry disease. I am looking forward to collaborating with expert clinicians in the field. These therapies could be game-changing.
Additionally, the findings of the suggested Italian paper have been included.
Line 524: You don't offer suggestions as to how leukocyte-mediated inflammation (which is likely different for neutrophils, B/T lymphocytes, monocytes, macrophages) then results in vasculopathy and damage at the tissue and organ levels. Also, why are autonomic and sensory neuropathies and GI symptoms manifest often in childhood in classical male FD whereas clinically manifest renal and cardiac damage delayed for several decades of life? You should devote some of the discussion to this instead of devoting so much verbiage to reiterating material that you presented earlier in the manuscript.
Response: I have incorporated a discussion on how different leukocyte populations, such as neutrophils, B and T lymphocytes, monocytes, and macrophages contribute to inflammation and the resulting vascular and tissue complications. Additionally, I have addressed the early onset of autonomic and sensory neuropathies, along with gastrointestinal symptoms, in childhood, juxtaposed against the delayed renal and cardiac damage that often manifests decades later. This distinction emphasizes the complex relationship between complement activation and immune responses in Fabry disease. Thank you for your valuable suggestion, which has allowed for a more comprehensive discussion of these important topics.
Line 527: How does this process occur
Response: The mechanism by which Gb3 induces complement activation and subsequently leads to the production of C3a and C5a has been thoroughly addressed. I have provided a detailed explanation of this process in the revised manuscript, along with a new Figure 2 to illustrate the pathways involved. This should enhance the understanding of how Gb3 contributes to the inflammatory response in Fabry disease. Thank you for your inquiry, which prompted a more in-depth exploration of this crucial aspect.
In a couple of places, you make reference to "in our study." Were you referring to this review article or to work that you have previously published?
Response: I appreciate your attention to clarity regarding the references to “in our study”. I intended to refer specifically to our previously published work rather than this review article. I ensure that this distinction is made clearer throughout the manuscript to avoid any confusion.
Comments on the Quality of English Language
I noticed several typos or grammatical errors that I highlighted in the uploaded file.
Response: Typos or grammatical errors have been corrected throughout the MS.
I believe that the revisions made in response to Reviewer 1 comments have significantly strengthened the manuscript. I am grateful for the opportunity to improve this work and hope that the revised version meets the expectations of the editorial team and the reviewers.
Sincerely,
Manoj Pandey
Reviewer 2 Report
Comments and Suggestions for Authors
I read with great interest your manuscript Complement System and Adhesion Molecule Skirmishes in Fabry Disease. Your study is an important contribution toward a better understanding of the complex pathogenesis and progression of tissue damage in Fabry disease and perhaps will give rise towards other therapeutic approaches of this disease.
As the endothelial dysfunction, the increased levels of complements C3a and C5a as well as enhanced expression of adhesion molecules on leucocytes and endothelial cells play a pivotal role in the pathogenesis of inflammation of tissues I suggest that you focus your manuscript on these compounds and, as is now the case, don't go into lots of detailed information on all immunological changes that are associated with the disease.
The accumulation of Gb3 in the lysosome and further on in tissues is central in Fabry disease. As Gb3 is mostly accumulating in cells and tissues the more water soluble compound lyso-Gb3 is found in serum/plasma: does this lyso-Gb3 play a role in the immunological dysfunctions that are described in your manuscript?
Although you refer to the work of the group of Lenders M. and Brand E. I miss a publication of Solvey Pollmann as first author on the destruction of the endothelial glycocalyx in Fabry patients. In my opinion your work and the glycocalyx story can be linked and are complementary in providing insight in the endothelial disruption giving rise to inflammation of the underlying tissues. Please comment on this.
Minor comments:
-you mention hyperhidrosis as a clincal sign of Fabry disease. Most patients, especially at a younger age however manifest hypohidrosis;
-Gb3=globotriaosylceramide; you should mention this once in your manuscript before using the abbreviation
Author Response
I sincerely appreciate the reviewer’s positive feedback and insightful recommendations, which have been invaluable in refining the manuscript's focus and clarity.
I want to address pointwise all the interests raised and provide clarifications accordingly.
Reviewer 2
I read with great interest your manuscript Complement System and Adhesion Molecule Skirmishes in Fabry Disease. Your study is an important contribution toward a better understanding of the complex pathogenesis and progression of tissue damage in Fabry disease and perhaps will give rise towards other therapeutic approaches of this disease.
As the endothelial dysfunction, the increased levels of complements C3a and C5a as well as enhanced expression of adhesion molecules on leucocytes and endothelial cells play a pivotal role in the pathogenesis of inflammation of tissues I suggest that you focus your manuscript on these compounds and, as is now the case, don't go into lots of detailed information on all immunological changes that are associated with the disease.
Response: I have taken the reviewer’s advice to emphasize the roles of complement components C3a and C5a, along with the enhanced expression of adhesion molecules on leukocytes and endothelial cells in the pathogenesis of tissue inflammation in Fabry disease. This streamlined focus highlights their central contributions to disease progression while still retaining some discussion of additional immunological aspects to provide a more comprehensive overview.
The accumulation of Gb3 in the lysosome and further on in tissues is central in Fabry disease. As Gb3 is mostly accumulating in cells and tissues the more water-soluble compound lyso-Gb3 is found in serum/plasma: does this lyso-Gb3 play a role in the immunological dysfunctions that are described in your manuscript?
Response: Regarding the query about the role of lyso-Gb3 in immunological dysfunctions, I have clarified that while lyso-Gb3 is present in serum due to Gb3 accumulation in tissues, complement activation specifically the production of C3a and C5a is primarily driven by Gb3. I referenced recent findings by Laffer et al., which show that elevated C3a concentrations in treatment-naive Fabry patients persist even after reductions in lyso-Gb3 levels, indicating that lyso-Gb3 may not significantly influence complement activation (PMCID: PMC10830671).
Although you refer to the work of the group of Lenders M. and Brand E. I miss the publication of Solvey Pollmann as first author on the destruction of the endothelial glycocalyx in Fabry patients. In my opinion your work and the glycocalyx story can be linked and are complementary in providing insight in the endothelial disruption giving rise to inflammation of the underlying tissues. Please comment on this.
Response: I appreciate the reviewer’s suggestion to go through the work of Solvey Pollmann regarding endothelial glycocalyx degradation in Fabry disease. I have included this important reference (PMID: 34917096) and elaborated on how our findings are linked to the glycocalyx narrative, providing insights into the mechanisms of endothelial disruption and the resultant tissue inflammation. This connection enriches the discussion and offers a more holistic view of the pathophysiology involved
Minor comments:
-you mention hyperhidrosis as a clincal sign of Fabry disease. Most patients, especially at a younger age however manifest hypohidrosis.
-Gb3=globotriaosylceramide; you should mention this once in your manuscript before using the abbreviation
Response: I have addressed the reviewer’s minor comments as follows.
The mention of hyperhidrosis has been corrected to reflect that most patients especially younger individuals, typically exhibit hypohidrosis.
The full term for Gb3 (globotriaosylceramide) has been defined upon its first use in the manuscript.
I believe that the revisions made in response to Reviewer 2 comments have significantly strengthened the manuscript. I am grateful for the opportunity to improve this work and hope that the revised version meets the expectations of the editorial team and the reviewers.
Sincerely,
Manoj Pandey